# The Role of Insulin-like Growth Factor I in Mechanisms of Resilience and Vulnerability to Sporadic Alzheimer’s Disease

**DOI:** 10.3390/ijms242216440

**Published:** 2023-11-17

**Authors:** Jonathan A. Zegarra-Valdivia, Jaime Pignatelli, Angel Nuñez, Ignacio Torres Aleman

**Affiliations:** 1Achucarro Basque Center for Neuroscience, 48940 Leioa, Spain; jonathan.zegarra@achucarro.org; 2Biomedical Research Networking Center on Neurodegenerative Diseases (CIBERNED), 28029 Madrid, Spain; jpigna@cajal.csic.es; 3School of Medicine, Universidad Señor de Sipán, Chiclayo 14000, Peru; 4Cajal Institute (CSIC), 28002 Madrid, Spain; 5Department of Anatomy, Histology and Neuroscience, Universidad Autónoma de Madrid, 28049 Madrid, Spain; angel.nunez@uam.es; 6Ikerbasque, Basque Foundation for Science, 48009 Bilbao, Spain

**Keywords:** Alzheimer’s disease, insulin-like growth factor I, vulnerability and resilience, risk factors

## Abstract

Despite decades of intense research, disease-modifying therapeutic approaches for Alzheimer’s disease (AD) are still very much needed. Apart from the extensively analyzed tau and amyloid pathological cascades, two promising avenues of research that may eventually identify new druggable targets for AD are based on a better understanding of the mechanisms of resilience and vulnerability to this condition. We argue that insulin-like growth factor I (IGF-I) activity in the brain provides a common substrate for the mechanisms of resilience and vulnerability to AD. We postulate that preserved brain IGF-I activity contributes to resilience to AD pathology as this growth factor intervenes in all the major pathological cascades considered to be involved in AD, including metabolic impairment, altered proteostasis, and inflammation, to name the three that are considered to be the most important ones. Conversely, disturbed IGF-I activity is found in many AD risk factors, such as old age, type 2 diabetes, imbalanced diet, sedentary life, sociality, stroke, stress, and low education, whereas the Apolipoprotein (Apo) E4 genotype and traumatic brain injury may also be influenced by brain IGF-I activity. Accordingly, IGF-I activity should be taken into consideration when analyzing these processes, while its preservation will predictably help prevent the progress of AD pathology. Thus, we need to define IGF-I activity in all these conditions and develop a means to preserve it. However, defining brain IGF-I activity cannot be solely based on humoral or tissue levels of this neurotrophic factor, and new functionally based assessments need to be developed.

## 1. Introduction

Mutations in amyloid precursor protein (APP) and preselinins are associated with familial Alzheimer’s disease (fAD), which constitutes around 1% of AD cases [1], and are the cause of overproduction of amyloid β (Aβ) peptides [2,3]. Excess production and reduced clearance of Aβ peptides have been postulated for many years as the major pathogenic pathway in AD [4]. fAD patients usually start to show symptoms at around the fourth decade of life [5], which suggests that overproduction of Aβ over the decades is required to start AD symptoms. In sporadic AD (sAD) associated with old age, it is now considered that Aβ accumulation starts at least 20 years before AD symptoms become evident, at around >65 years of age [6]. Hence, apparently, fewer years of Aβ accumulation are required in sAD to develop symptoms, compared to fAD. Among several potential explanations, it might be that a younger brain combats Aβ accumulation more efficiently than an older brain, as specific mechanisms of resilience to cognitive deterioration have been described [7,8], which may weaken along with age. Alternatively, it is possible that in sAD, other pathogenic pathways contribute to Aβ accumulation to reach a pathological threshold earlier. This threshold is specific for each individual [9], and a sizable proportion of elders (around 30%) show Aβ accumulation without AD symptoms [10]. As sAD is considered a multifactorial disease resulting from genetic/environmental interactions [11], while the former are, at present, difficult to overcome, environmental risk factors are possible to curtail. Indeed, lifestyle interventions are now implemented in personalized medicine protocols for AD patients [12], and constitute the basis of current therapeutic proposals [13] apart from pharmacotherapy.

Based on our observations and those from many other laboratories, we postulate that among other processes involved in the transition from prodromic to fully fledged sAD, such as vascular, metabolic, or immune disturbances [14], an additional underlying process that may help explain why a given lifestyle factor modulates sAD risk is brain insulin-like growth factor I (IGF-I) activity. We already postulated that loss of IGF-I function due to reduced IGF-I receptor sensitivity, IGF-I deficiency, or both, favors the development of pathogenic events related to AD [15]. In turn, AD pathology may also contribute to disrupted IGF-I activity. We later discussed in more detail the role of insulin peptides (ILPs) in the link between lifestyle and sAD risk [16], providing a mechanistic framework for this connection. Namely, inflammation, oxidative stress, altered proteostasis, impaired Aβ clearance, tau hyperphosphorylation, disturbed metabolism, and reduced cytoprotection would be processes resulting from impaired ILP activity. We now elaborate further this proposal, postulating a specific link between IGF-I activity and mechanisms of resilience and vulnerability in AD.

Many clinical studies link altered IGF-I activity and sAD risk (see, for example, [17,18]), but no definitive connection has been yet confirmed. We now propose that most of the established risk factors for sAD, including old age, type 2 diabetes, sedentary life, loneliness, unbalanced diets, stroke, and post-traumatic stress disorders (PTSD) are associated with impaired IGF-I function. Other well-established risk factors such as ApoE4 genotype may also be related to IGF-I dysfunction, although the evidence is not as robust. Traumatic brain injury (TBI), which also disturbs brain IGF-I activity, may also be a risk factor for sAD, but the evidence in this regard is not yet firm. At any rate, not only we [19] but also others already suggested a link between altered IGF-I activity and diverse AD co-morbidities and risk factors [20,21]. However, we would like to point out that other disturbances, besides altered IGF-I activity, will likely be involved in the connection between the above-mentioned risk factors and AD.

This growth factor shows a wide repertoire of actions in the brain, and its alteration will impact on many aspects known to be affected in AD. For a more detailed discussion of this topic, we refer readers to a previous review [16]. In brief, neurotrophic activity of IGF-I involves adult neurogenesis [22], re-innervation after insult [23], reduction in of inflammation [24] and oxidative stress [25], promotion of glucose uptake [26], and many others [27]. Its pro-cognitive actions include numerous effects on neuronal plasticity [28], cognition [29], and mood [30]. IGF-I is also involved in key homeostatic processes, including energy allocation [31] and the sleep/wake cycle [32]. The circadian activity of IGF-I [33] is likely involved in the latter. Table 1 shows a summary of major IGF-I activities in relation to AD and supporting references.

## 2. IGF-I and AD Resilience

The concept of AD resilience has been coined to explain the presence of AD pathology in cognitively intact individuals [39]. The specific mechanism underlying AD resilience is still undetermined and is often related to the concept of cognitive reserve (see below). Resilience to AD seems in part to be genetically determined as it shows a sex-dependent inheritable architecture [40], and this is not surprising considering the heavy genetic make-up of AD risk [41]. This genetic component may help uncover novel targets of resilience, such as the recently reported reelin, a protein functionally related to ApoE [42], a major genetic risk factor for sAD. However, the bulk of mechanisms of AD resilience are not genetic, and novel proposals are needed.

Accordingly, several lines of research are trying to shed light on AD resilience, as it appears very promising to develop novel routes of AD therapy. For example, early life context [43], aerobic glycolysis [44], efficient microglial phagocytosis [45], and dendritic spine plasticity [46] have all been suggested to contribute to resilience/vulnerability to AD. Therefore, understanding the underlying mechanisms will unveil new potential targets in AD prevention. In this vein, while no general consensus has yet been reached, and the major conclusions indicate that further work is needed to firmly establish a causal link between circulating IGF-I levels and cognition [47], available information allows us to suggest that preserved brain IGF-I activity also contributes to resilience to AD pathology. Thus, all the major characteristics found in individuals resilient to AD can be explained in the light of preserved brain IGF-I activity. These include conserved neuronal numbers, synaptic markers, and axonal architecture, as well as cytokine profiles consisting of higher anti-inflammatory signals and neurotrophic factors, and lower cytokine mediators of microglial recruitment [48,49]. Indeed, recent ideas supporting a multifactorial approach to treating cognitive loss in dementia [13] can be accommodated in our proposal if we consider these multi-pronged measures as a means to preserve brain IGF-I activity, such as through behavior (Figure 1).

### Mechanisms of IGF-I-Dependent AD Resilience

Potentiation of neurotrophic activity, most often BDNF [51], has already been invoked as a mechanism of AD resilience [52], but specific mechanisms and factors need to be defined. Since the neuroprotective actions of IGF-I are pleiotropic [27,47], all the major characteristics found in AD resilience can be readily explained through them. These variety of IGF-I effects involve different pathways, as explained in detail elsewhere [16]. Importantly, other neurotrophic pleiotropic factors, such as melatonin, have also been implicated in AD resilience through longevity signals, such as Sirt1, or anti-inflammatory pathways involving NFκB [53]. Therefore, it is very likely that different neurotrophic activities, and not only IGF-I, are involved in resilience to AD.

As for the mechanisms underlying IGF-I-mediated AD resilience, we first focus on cell-based processes that affect all types of brain cells [54]. Among them, synapse loss is considered a major structural disturbance associated with cognitive deterioration in sAD [55]. Thus, IGF-I is involved in physiological synaptogenesis during development [56], in adult brains [57], and in synapse repletion after an insult [58]. Importantly, dendritic spines, a major site of cortical synapses, provide AD resilience [46], while IGF-I promotes dendritogenesis [59] and is intricately involved in synaptic physiology [60,61].

Another process that is emerging as an important event in cellular changes in AD is neuro-inflammation, classically associated with astrocytes and microglia as the main cellular effectors [62,63]. We must remember that inflammation is primarily a homeostatic response to pathology, and when it becomes maladaptive, for as yet poorly described reasons, it constitutes a key factor in driving sAD pathology [64,65], leading to the alteration of structural and functional brain networks seen in AD, as recently reported [66]. This “double-edge sword” process [67] is also modulated by IGF-I acting through a calcineurin-NFκB pathway in astrocytes that reversibly drives AD pathology in AD mice [24]. Naturally, neuro-inflammation also impacts on many other cellular activities, such as astrocyte phagocytosis [68], microglial reactivity [69] and proliferation [70], and activity of brain resident macrophages [71], and it also interacts with the brain angiotensin anti-inflammatory pathway [72,73]. The involvement of IGF-I in the response to neuro-inflammatory processes associated with brain damage in general attests to an important role of IGF-I in neuro-inflammation [72]. Conversely, neuro-inflammation associated with AD will contribute to IGF-I resistance in a “vicious circle” often described in the AD pathological cascade.

Other cell-associated processes in AD pathology, such as excess oxidative stress [74], which is probably directly involved in AD-related cell demise [75], are also counteracted by IGF-I [25]. Since an efficient mechanism of prevention of oxidative stress has been suggested to work in the brain of individuals showing AD resilience [76], antioxidant actions of IGF-I in brain tissue could be forming part of this resilience. Moreover, tau hyperphosphorylation in neurons, a hallmark of AD, can also be ameliorated by IGF-I through its capacity to inhibit tau kinases such as GSK-3 [77]. Indeed, IGF-I null mice show brain tau hyperphosphorylation [78]. Finally, disturbed proteostasis, a common trait in many neurodegenerative diseases [79] and considered a major culprit in AD [80], is also related to brain IGF-I actions affecting brain Aβ clearance, catabolism, and neurotoxicity [81,82,83].

At the system level, dysregulated neural circuit activity [84,85] and an altered astrocytic network [86,87], or both disturbances interacting with each other [88,89], are postulated to participate in the initiation and maintenance of the AD pathogenic cascade. While diverse explanations have been proposed, including early alterations of peptidergic systems [90,91], tau accumulation [92], or early loss of inhibitory tone [93], impaired brain IGF-I activity may also be involved. Although the evidence is less robust than its relation to cell-based processes related to AD pathology, it is well documented that IGF-I regulates neuronal activity at various levels. Thus, IGF-I modulates neuronal excitability [60] and excitatory/inhibitory balance [29,94], which also includes its actions through astrocytes [37], a type of glial cell known to modulate neuronal circuits. Indeed, we recently argued that regulation of neuronal activity by IGF-I is so widespread that it may underlie its role as an interoceptive neuromodulatory signal involved in brain states [47].

Another system-level disturbance associated with AD is insulin resistance, as seen in type 2 diabetes [95]. In this case, the evidence linking IGF-I activity with insulin sensitivity and brain insulin actions is robust [96,97], even though the hierarchical structure of these relationships is not yet defined. Finally, a vascular-related disturbance underlying sAD pathology was invoked decades ago, and is of potential relevance to this disease, if only because vascular disturbances are commonly associated with AD pathological hallmarks [98]. Again, brain IGF-I is instrumental in brain vascular function [99].

Both higher education and physical activity are associated to better mental health and are claimed to promote resilience to AD [100,101,102,103]. Thus, a straightforward connection between preserved brain IGF-I activity and AD resilience can be established. Indeed, both increased mental activity associated with environmental enrichment [104] and higher physical activity associated with exercise promote brain IGF-I function [105,106].

The aforementioned processes illustrate the pleiotropic actions of IGF-I on brain cells since multiple aspects of cell physiology appear to be targeted by IGF-I. This is a key characteristic of brain IGF-I function that probably is present in other organs.

## 3. IGF-I and AD Risk

We will discuss now those instances where disturbed brain IGF-I activity likely helps explain its association with AD risk. The underlying mechanisms usually relate to reduced IGF-I activity, resistance to IGF-I actions, or even both.

### 3.1. Old Age

Age is the most important risk factor for sAD [1] and is associated with a decline in the activity of IGF-I [107] in the form of deficiency [108] and resistance [109], affecting also the brain [110]. This reduced activity is sufficient to explain the lower IGF-I-dependent resources to combat age-associated deleterious changes that may contribute to the development of the pathogenic cascade in AD. Reduced IGF-I input during aging compromises health span in general [111], while in the brain, this deficiency impacts on vascular function [112], neuro-vascular coupling [113], cognition [114], mood balance [115], and sensory perception [116,117]. A particular characteristic of aging that is presently gaining attention in its relation to AD pathology is that it is frequently associated with disturbed circadian rhythms in the brain and in peripheral organs [118], mostly because AD patients show altered sleep/wake timing [119]. Of note, Aβ clearance takes place mostly during sleep [120], while IGF-I regulates the expression of circadian clock genes [121], and its production depends on them [33]. Accordingly, we recently observed that IGF-I also modulates the sleep/wake cycle [122] and other circadian behaviors [123].

Despite an obvious association between reduced IGF-I activity during aging and deleterious changes in brain function found in sAD, current mainstream thinking poses brain IGF-I activity as detrimental in AD pathology [124,125]. Thus, whether age-associated lowering IGF-I activity is adaptive or maladaptive for AD, along with other brain maladies [126,127], is still debated. We recently addressed this controversy, favoring the proposal that the IGF-I receptor (IGF-IR) is a dependence receptor [128] with ligand-independent actions that are counter-regulated by IGF-I [129]. We reasoned that in old age, IGF-I activity is decreased, and ligand-independent actions of IGF-IR remains unchecked. However, more work is needed to clarify the potential causal role of IGF-I in AD pathology.

### 3.2. Type 2 Diabetes

Loss of insulin sensitivity underlying type 2 diabetes (T2D) is linked to age [130] and affects also IGF-I sensitivity [131], as both ILPs are functionally interrelated in the control of glucose handling [31,132] and probably in other functions [128]. Since different lines of evidence support that T2D is a risk factor for sAD [133], in recent years, a causal connection between T2D and AD has been extensively explored and discussed [95,134], although no firm conclusions have been reached yet. Thus, it is not clear whether T2D favors AD [135,136], or vice versa [137,138], or whether both conditions evolve in parallel [139]. Among the favored mechanisms underlying this connection, T2D-induced brain Aβ accumulation [140], tau hyperphosphorylation [141], Aβ-induced loss of insulin sensitivity [142], T2D-associated oxidative stress and inflammation [134], subclinical blood–brain barrier (BBB) breakdown [143], and defective insulin signaling [144] have been proposed.

We consider that brain disturbances associated with T2D, most prominently cognitive deterioration [145], may be related not only to the underlying metabolic alterations but also to a dysfunctional ILP system (which includes IGF-binding proteins) that will interfere with essential homeostasis processes, such as central control of energy allocation [97,146], inflammation/oxidative stress, feeding, or the sleep/wake cycle, as discussed above.

### 3.3. Imbalanced Diet

A link between diet and AD has long been recognized in epidemiological studies [147], although recent evidence questions a direct cause–effect relationship since diverse contradictory observations [148,149] no longer allow us to unequivocally ascribe obesity as a risk factor for AD, although a majority of analyses favor this connection [150]. As type 2 diabetes, metabolic syndrome, and imbalanced diets are closely linked, metabolic alterations associated with improper feeding behavior still need to be considered important contributors in the path to AD, as pilot experiments have suggested [151]. At any rate, diet habits long considered to be protective, such as the Mediterranean diet, are probably beneficial in attenuating AD risk [152]. Among the potential pathogenic drivers in the link between diet and AD, neuroinflammation is considered the best candidate [153] as it is a disturbance associated with reduced IGF-I activity [154].

### 3.4. Sedentary Life

Among the modifiable lifestyle factors included in preventive schemes for sAD and many other maladies, an active life is commonly considered. There is now ample experimental and epidemiological evidence that physical activity is an effective measure to preserve cognitive abilities [155], with therapeutic application [156]. Experimental evidence suggests that the therapeutic efficacy of physical activity differs from that provided by mental activity [157]. However, this distinction may not be relevant for our argument, as it seems that brain activity per se [105], regardless of what triggers it, stimulates brain uptake of circulating IGF-I. Conversely, since serum IGF-I is a mediator of exercise neuroprotection [158], we suggested that disturbed IGF-I action in the brain contributes to the deleterious effects of a sedentary life through a loss of homeostatic repair mechanisms [159].

### 3.5. Low Education

Years in school are inversely associated with the risk of dementia [160] and rate of cognitive decline [161]. The concept of cognitive reserve [162], with more functional resources available, provides a commonly used explanation for this link [163]. This concept helps interpret the fact that around 30% of people with AD pathology show normal cognition [164], with explanations varying from larger brains or increased neuronal plasticity in individuals with higher mental activity linked to educational status. While attempts to identify the processes purportedly connecting cognitive reserve and sAD risk are still not satisfactory [165], mendelian randomization analyses point to cognitive performance associated with cognitive reserve as a direct cause of protection against AD [166]. Significantly, both brain growth [167] and neuronal plasticity are directly related to brain IGF-I activity [168]. From our point of view, higher education is associated with sustained higher mental activity, which will preserve brain IGF-I activity.

### 3.6. Stroke

Cerebrovascular accidents are considered the “silent pandemic” as they are the second cause of death worldwide [169]. Unfortunately, they are widely held as an sAD risk factor [170], and vascular pathology is the most common co-morbidity observed in AD brains [98]. Among the potential mechanistic links between stroke and sAD, defects in interstitial fluid drainage of Aβ peptides [171], development of cerebral amyloid angiopathy [172], tissue hypoxia [173], and excitotoxicity [174] have all been argued as pathological disturbances resulting from stroke that favor the development of sAD. In line with our proposal that IGF-I activity is responsible for the link between stroke and sAD, IGF-I activity in stroke patients is altered [175], while disturbed IGF-I activity underlies cerebrovascular dysfunction in AD mice [176]. In addition, insulin resistance, a consequence of IGF-I dysfunction [132,177], also links stroke with AD [178].

### 3.7. Post-Traumatic Stress Disorder

This condition is another instance that may help explain the increasing world incidence of sAD. Hence, post-traumatic stress disorder (PTSD) also shows an increased incidence worldwide, being linked to conflicts, natural disasters, and climate change [179], and may also be a risk factor for sAD, although this is still not firmly settled yet [180,181]. This is not surprising as stress in general is linked to AD pathology [182]. In this vein, as low serum IGF-I is linked to higher vulnerability to stress in humans and mice [30], and vulnerability to stress is also linked to AD risk [183], a direct link between stress and AD risk is readily justified by low IGF-I activity.

Notwithstanding a link between stress in general, IGF-I, and AD, in the particular case of PTSD, sleep disturbances associated with this condition have been argued to have a link with AD [184]. However, no AD-like pathology is found in PTSD patients [185]. Since sleep is associated with Aβ clearance [120], sleep disturbances are found in mice with reduced IGF-I activity in hypothalamic orexin neurons [32], and these mice develop PTSD-like features upon exposure to trauma [94], we propose that the link between PTSD and AD is mediated by faulty IGF-I activity in the hypothalamus. At any rate, PTSD exacerbates AD pathology in mouse models [186].

We next discuss those factors with weaker evidence that brain IGF-I activity underlies their connection with sAD risk.

### 3.8. ApoE4

The E4 allele of ApoE is a major genetic contributor to sAD [187]. Only a few scattered observations link ApoE4 with IGF-I, with both seemingly interacting with each other. Therefore, no robust evidence is yet available regarding a role of IGF-I in the genetic risk posed by ApoE4. Higher levels of serum IGF-I are associated with the ApoE4 genotype in a UK BioBank sample of 400,000 individuals [188], while a modifying role of ApoE4 has been ascribed to the connection between serum IGF-I levels and brain network activity in a >13,000 large cohort of the UK Biobank [189]. It has also been pointed out a modifying effect of IGF-I polymorphisms on the genetic risk of AD [190], whereas the ApoE4 genotype modulates brain responses to insulin [191], whose effects are related to IGF-I. Further, the ApoE genotype modulates the brain IGF system [192].

### 3.9. Traumatic Brain Injury

An association of traumatic brain injury (TBI) with a greater risk of developing dementia has also been postulated, and TBI is considered a risk factor for sAD [193], although there is still no general agreement [194]. Increased Aβ production early after TBI [195], supported by clinical observations [196], provides an easy explanation for the underlying IGF-I dysfunction, as IGF-I has been postulated to participate in Aβ clearance [81]. Indeed, a previous proposal already suggested that IGF-I underlies the link between TBI and AD [197], and recent data tend to substantiate this claim as many of the newly reported alterations purportedly linking both conditions may involve IGF-I. Thus, the resultant vascular injuries [198], axon damage favoring Aβ and hyperphosphorylated tau accumulation [199], or neuro-inflammation [200] may readily associate with IGF-I dysfunction. However, not all the changes recently reported, such as enhanced production of acetylated tau [201] or TDP-43 [202]; disturbed cathepsin B [203] or delta-secretase function [204]; and specific post-TBI disturbances identified in patients such as diffuse axonal injury, which provide an explanation for the neurodegenerative changes probably anteceding sAD [205], can yet be associated with IGF-I.

## 4. Outlook

Defining what we could consider a preserved IGF-I activity is challenging. A range of serum IGF-I values in the normal population has not yet been recorded since technical shortcomings have not been entirely solved [206,207,208]. Further, due to the complex biology of the IGF system, which includes IGF-I and -II together with six binding proteins, serum concentrations are, in all likelihood, insufficient to define IGF-I activity in target organs. In an attempt to respond to this limitation, clinical assays, such as “free IGF-I” [209] or “IGF-I binding activity” [210], have been proposed, but their clinical validity is not yet confirmed at a general level. Therefore, this first goal is still distant. Moreover, for a better functional account of the actual IGF-I input to the brain, we suggest that other approaches should be implemented. As an example, we already proposed [211] an exercise-based procedure to probe IGF-I activity in the brain by combining a bout of exercise (similar to procedures already used in the clinical setting to determine cardiovascular fitness) with electrophysiological (EEG) recordings of brain activity (Figure 2). This protocol is intended to be used in clinical practice and is based on previous findings of exercise-induced IGF-I entrance into the brain [106] and IGF-I-induced changes in EEG patterns [212].

## 5. Summary

The shortcomings of current concepts in AD research are now widely accepted [213]. Among the elegant proposals [214] and elaborated suggestions [213] formulated to provide a working framework to elucidate the etiopathogenic processes in sAD, we put forward a reductionist approach for furthering research into the development of novel AD therapies based on the notion that IGF-I activity in the brain may serve as a biomarker of resilience/vulnerability to AD pathology. Although faulty brain IGF-I activity will not be the sole factor leading to a multifactorial, highly complex disease such as sAD, we propose that it should be taken into account as a potential therapeutic aid in conjunction with novel avenues of treatment.

## Figures and Tables

**Figure 1 ijms-24-16440-f001:**
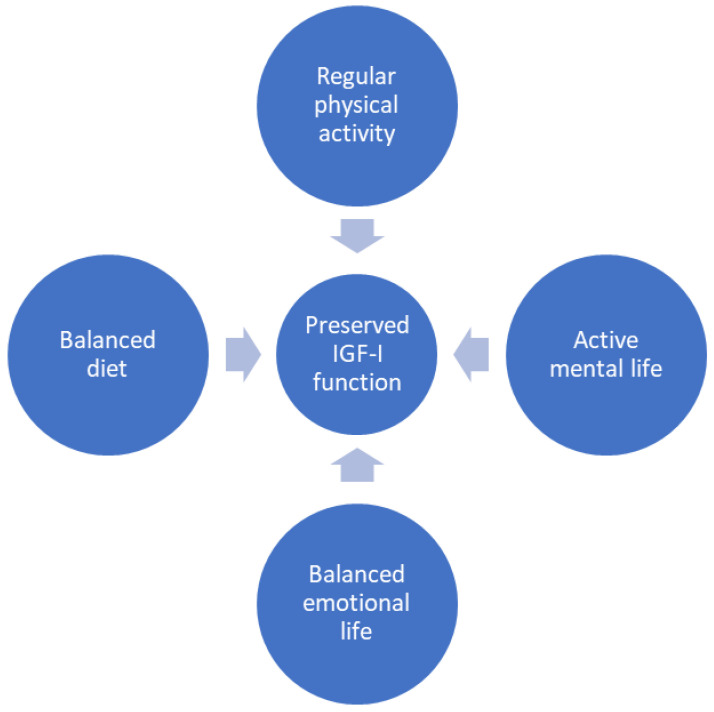
Preservation of brain IGF-I activity through behavior. There are four main behavioral approaches to preserving brain IGF-I activity. Three of them, balanced emotional life and active physical and mental life, are already well established to reduce sAD risk. Reduction in stress associated with current lifestyle through different approaches (i.e., meditation), regular (moderate) exercise, and engagement in cognitively demanding tasks, including social intercourse, are becoming common knowledge in the prevention of sAD. In the case of balanced diets, numerous studies have not yet reached a firm consensus for any one in particular, although the Mediterranean diet is probably the most favored at present [50]. Modified from [16].

**Figure 2 ijms-24-16440-f002:**
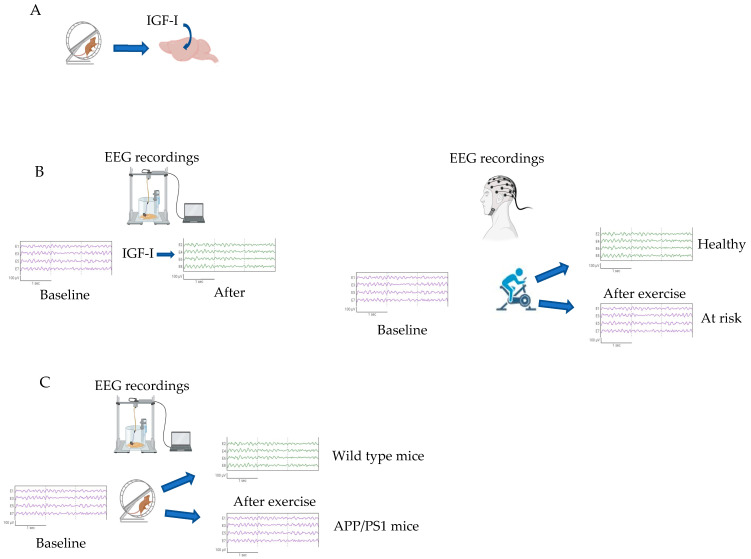
Indirect determination of IGF-I activity in the brain using electro-encephalographic (EEG) responses to exercise as a surrogate. The proposed test is based on three observations in mice: (**A**) Exercise induces the entrance of circulating IGF-I into the brain [106]. (**B**) Circulating IGF-I modulates EEG patterns [212]. (**C**) Absence of exercise-induced changes in EEG patterns in mildly cognitively impaired APP/PS1 mice [211]. Right panel: the proposed test for measuring brain IGF-I activity in humans consists of EEG recordings at rest, followed by a bout of moderate exercise and subsequent EEG recordings. The prediction is that subjects showing no EEG changes after exercise are at risk of developing cognitive disturbances. The purported explanation is that IGF-I activity is lower in the brains of these subjects.

**Table 1 ijms-24-16440-t001:** Reported IGF-I activities related to mechanisms known to be altered in AD.

Activity	Main References
Neurogenesis	[22,34]
Re-innervation	[23]
Regulation of inflammation	[24,35]
Regulation of oxidative stress	[25,36]
Neuronal plasticity	[28,37]
Cognition	[29]
Mood	[30]
Energy allocation	[26,31,38]
Sleep/wake cycle	[32]

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
