# Peer review of "The Role of Insulin-like Growth Factor I in Mechanisms of Resilience and Vulnerability to Sporadic Alzheimer’s Disease"

_ijms, 2023, doi:10.3390/ijms242216440_

Round 1

Reviewer 1 Report

Comments and Suggestions for Authors

The present review article by Zegarra-Valdivia el al deals with the impact of the IGF1 system on mechanisms of resilience and vulnerability to sporadic Alzheimer’s disease (AD). Authors evaluate the hypothesis that IGF1 activity in the brain provides a common substrate for mechanisms of resilience and vulnerability to AD. Furthermore, they postulate that preserved brain IGF1 activity provides resilience to AD pathology, as the IGF1 axis participates in a number of pathological cascades that are regarded as critical in AD etiology.

This is an extremely well-written, very didactic and updated paper. Authors discuss the mechanisms of IGF1-dependent AD resilience, including neuro-inflammation, oxidative stress, and others. In addition, they describe a number of risk factors associated with disruption of IGF1 function. These risk factors include: age, type 2 diabetes, imbalanced diet, sedentary life, etc. Finally, authors suggest that IGF1 activity in the brain may serve as a biomarker of resilience/vulnerability to AD pathology.

Specific points:

Line 109: should be “…all the major characteristics found in …”.

Lines 162-166: the sentence starting “While the evidence is less robust…” is very long, confusing and grammatically wrong. Please divide into two sentences.

Line 165: should be “… these cells modulate…”.

Line 167: remove comma after “widespread”.

Lines 176-177: should be “…and physical activities …”.

Line 252: Authors stated “Years at school are directly associated to risk of dementia”. Is it ‘directly’ or ‘inversely’?

References 42 and 79 are incomplete.

Author Response

Specific points:

Line 109: should be “…all the major characteristics found in …”.

Answer: corrected

Lines 162-166: the sentence starting “While the evidence is less robust…” is very long, confusing and grammatically wrong. Please divide into two sentences.

Answer: The sentence (in ln 144-147) has been divided into two.

Line 165: should be “… these cells modulate…”.

Answer: corrected (in ln 147)

Line 167: remove comma after “widespread”.

Answer: removed (in ln 148)

Lines 176-177: should be “…and physical activities …”.

Answer: corrected (in line 156)

Line 252: Authors stated “Years at school are directly associated to risk of dementia”. Is it ‘directly’ or ‘inversely’?

Answer: changed for inversely (in line 225)

References 42 and 79 are incomplete.

Answer: Amended

Reviewer 2 Report

Comments and Suggestions for Authors

The current paper sets out to link IGF-I and all aspects of sporadic AD. The idea is interesting but probably a little overstated. The author cannot prove that other changes are changing IGF-I. Certainly IGF-I changes with aging and disease whether it is causal is not proven here. 

The "Sociality" section should be removed. 

Lines 63-64 and 75-76 are redundant. 

Please see attached for additional comments.

Author Response

The current paper sets out to link IGF-I and all aspects of sporadic AD. The idea is interesting but probably a little overstated. The author cannot prove that other changes are changing IGF-I. Certainly IGF-I changes with aging and disease whether it is causal is not proven here. 

Answer: Thank you for the comment. We tried not to be over-assertive, but it seems we failed, sorry!. We have now nuanced our argument that changes in IGF-I activity associated to aging and/or disease contribute to AD pathology. Please see changes in the Abstract, ln 129-132, and ln 280.

Lines 63-64 and 75-76 are redundant

Answer: The sentence in former lines 75-76 has been deleted to avoid redundancy.

I do not believe that the effect of IGF-1 and sociality was supported and would best be removed.  

Answer: Section removed.

They need to acknowledge that other factors in AD could be changing IGF-1 and not just the other way around.

Answer: We have introduced sentences addressing this bi-lateral relationship: ln 60 and ln 149-151.

It is important but not necessarily causal for all of the things they claim.

Answer: We have nuanced our hypothesis, see ln 74-76.

In terms of neurodegeneration, it is a biomarker, but more research is needed to show causal. The paper does point out that it should be examined. They need to point out that more research is needed to be definitive. 

Answer: Additional comments have been added: ln 208-209 and ln 329.

It is interesting. 

Answer: Thank you for the positive opinion.